# Coupled exoskeleton assistance simplifies control and maintains metabolic benefits: A simulation study

Nicholas A. Bianco[1]*, Patrick W. Franks[1], Jennifer L. Hicks[2], Scott L. Delp[1,2,3]

**1** Department of Mechanical Engineering, Stanford University, Stanford, California, United States of America,
**2** Department of Bioengineering, Stanford University, Stanford, California, United States of America,
**3** Department of Orthopaedic Surgery, Stanford University, Stanford, California, United States of America

* nbianco@stanford.edu

**Data Availability Statement:** All walking data and result files from this study are available at https://simtk.org/projects/coupled-exo-sim, and the source code used to generate the results are

## Abstract

Assistive exoskeletons can reduce the metabolic cost of walking, and recent advances in exoskeleton device design and control have resulted in large metabolic savings. Most exoskeleton devices provide assistance at either the ankle or hip. Exoskeletons that assist multiple joints have the potential to provide greater metabolic savings, but can require many actuators and complicated controllers, making it difficult to design effective assistance. Coupled assistance, when two or more joints are assisted using one actuator or control signal, could reduce control dimensionality while retaining metabolic benefits. However, it is unknown which combinations of assisted joints are most promising and if there are negative consequences associated with coupled assistance. Since designing assistance with human experiments is expensive and time-consuming, we used musculoskeletal simulation to evaluate metabolic savings from multi-joint assistance and identify promising joint combinations. We generated 2D muscle-driven simulations of walking while simultaneously optimizing control strategies for simulated lower-limb exoskeleton assistive devices to minimize metabolic cost. Each device provided assistance either at a single joint or at multiple joints using massless, ideal actuators. To assess if control could be simplified for multi-joint exoskeletons, we simulated different control strategies in which the torque provided at each joint was either controlled independently or coupled between joints. We compared the predicted optimal torque profiles and changes in muscle and total metabolic power consumption across the single joint and multi-joint assistance strategies. We found multi-joint devices–whether independent or coupled–provided 50% greater metabolic savings than single joint devices. The coupled multi-joint devices were able to achieve most of the metabolic savings produced by independently-controlled multi-joint devices. Our results indicate that device designers could simplify multi-joint exoskeleton designs by reducing the number of torque control parameters through coupling, while still maintaining large reductions in metabolic cost.

hosted at https://github.com/stanfordnmbl/coupled-exo-sim.

**Funding:** NAB, JLH, and SLD received support from the National Institutes of Health (https://www.nih.gov) grants U54 EB020405, P2C HD065690, P2C HD101913, and P41 EB027060. NAB received support from the National Science Foundation Graduate Research Fellowship Program and the Stanford Graduate Fellowship Program (https://vpge.stanford.edu/fellowships-funding/sgf). PWF received support from the U.S. Army Natick Soldier Research, Development and Engineering Center, grant W911QY18C0140. The funders had no role in study design, data collection and analysis, decision to publish, or preparation of the manuscript.

**Competing interests:** The authors have declared that no competing interests exist.

# Introduction

Wearable robotic exoskeletons that reduce the metabolic cost of walking could improve mobility for many individuals including those with musculoskeletal or neurological impairments and assist soldiers and firefighters carrying heavy loads. Assistance strategies that reduce metabolic cost have only recently been discovered using both powered [1–4] and unpowered [5] devices. Despite these successes, designing controllers for exoskeletons can be counterintuitive and time-consuming. Some exoskeleton designs focused on biomimicry, where assistive devices attempt to emulate biological joint kinematics, kinetics, and power, but these seemingly intuitive approaches have had limited success in reducing metabolic cost [6, 7]. To better understand what aspects of exoskeleton assistance affect metabolic cost, many recent studies have designed assistance by varying the timing and magnitude of assistive torques and powers [8–12]. For example, a recent study showed that optimizing both assistance torque onset timing and average power together produces larger metabolic reductions than when considering each quantity alone [11]. More recent approaches, such as human-in-the-loop optimization experiments, which continuously optimize assistance for a subject based on real-time estimates of metabolic energy, have produced large reductions in metabolic cost [8, 10]. However, since each human-in-the-loop optimization evaluation requires several minutes of human metabolic data from indirect calorimetry, it is time-consuming and expensive to test a large number of devices. For example, a human-in-the-loop optimization may take several days of experimentation to complete.

Simulations and experiments suggest that assisting multiple joints at once could deliver larger metabolic savings than assisting a single joint [12–15]. However, designing assistance for these "multi-joint" exoskeletons can magnify the challenges of optimizing the control, since such devices can include multiple actuators with independent control laws, which increases the number of parameters that must be tested in experiments. For example, the convergence time for human-in-the-loop optimization experiments scales poorly with increasing optimization variables, and therefore may be prohibitively long for multi-joint exoskeletons due to the large number of control variables needed for several assistive torques. As a result, most exoskeleton studies focus on assisting only one degree of freedom to simplify parameter design, usually preferring the hip or the ankle since they produce most of the positive power during walking and running [4, 16–18].

Coupled assistance could greatly simplify the mechanical and control design of exoskeleton devices by reducing control complexity (i.e., the number of parameters personalized to a subject) and thus reducing the time needed to perform human-in-the-loop optimizations to achieve desired reductions in metabolic cost. Coupled assistance could also simplify the mechanical design of exoskeletons which could make the device lighter and less restrictive for the sure. Assisting two joints at once using one actuator, or "coupling" assistance, produced significant reductions in metabolic cost in recent exoskeleton studies with an ankle-hip soft exosuit [12, 19–21] and a knee-ankle device [14]. These studies exploit the similar timings of joint moments (e.g., the hip flexion moment and ankle plantarflexion moment reach a maximum at approximately the same point in the gait cycle). Other exoskeletons that assist multiple joints may be effective, but they have not been tested in experiments. Simulations could help us identify which combinations of joints to assist and how control could be coupled across joints, while achieving significant decreases in metabolic cost.

Musculoskeletal simulation has become a valuable tool for examining the complex muscle-level and whole-body metabolic changes produced by exoskeleton devices [22]. Researchers have used simulation to analyze an existing exoskeleton and optimize its mechanical design [23] and to better understand human-device interaction [24]. Other studies have used

simulation to help interpret experimental results, for example, to understand how muscle mechanics drive metabolic changes for an ankle exoskeleton [25]. Researchers have also used simulation to model exoskeleton devices as ideal actuators to discover guidelines for designing walking [26] and running [13] exoskeletons. A recent study [27] applied results from assisted running simulations [13] to design assistance for a soft running exoskeleton. The simulation-derived controls provided greater metabolic cost reductions compared to assistance designed based on biological joint moments, demonstrating the ability of simulations to improve exoskeleton design. Another recent study conducted by our group used simulation to design assistance for an experimental hip-knee-ankle exoskeleton, resulting in a large metabolic reduction [28]. While this study and the running simulation study examined multi-joint assistance [13], no study has used simulation to systematically compare different multi-joint assistance strategies for walking.

In this study, we examined how simulated multi-joint assistance affects the metabolic cost of walking. We added ideal, massless assistive devices to a lower-extremity musculoskeletal model and simultaneously optimized muscle activity and device controls to match the net joint moments of normal walking and minimize metabolic cost. Each device assisted a single joint or assisted multiple joints simultaneously. Multi-joint devices could control assistance at joints independently or couple assistance for multiple joints, using the same control with independent peak torque magnitudes. We used the simulations to achieve two goals. First, we sought to estimate the metabolic savings provided by multi-joint exoskeletons during walking as compared to exoskeletons that assist only a single joint. Second, we sought to determine if coupled assistance could achieve similar metabolic savings compared to independent assistance. To address our second aim, we compared total and muscle metabolic cost savings and optimal device torques between coupled and independent multi-joint assistance.

## Materials and methods

### Experimental data

We used a previously-collected dataset from 5 healthy individuals walking on a treadmill (mean ± s.d.: age: 29.2 ± 6.3 years, height: 1.80 ± 0.03 m, mass: 72.4 ± 5.7 kg) [29]. Subjects in this previous study provided informed consent to a protocol approved by the Stanford Institutional Review Board. The data included marker trajectories, ground reaction forces, and electromyography (EMG) signals. For each subject, we simulated three gait cycles of walking at 1.25 m/s. One gait cycle was used in a model calibration step, and the other two were used for simulations of exoskeleton devices. For validating muscle activation patterns predicted from simulation, we used the processed EMG signals as described in the previous study [29], where signals were normalized by the highest value recorded across all walking speeds (see section "Comparison of simulations with experimental results").

### Musculoskeletal model

A generic 29 degree-of-freedom skeletal model was scaled to each subject's data based on static marker trials [30]. Nine Hill-type muscle-tendon units, as modeled in a previous simulation study from our group [31], were included on each leg of the model: gluteus maximus, biarticular hamstrings, iliopsoas, rectus femoris, vasti, biceps femoris short head, gastrocnemius, soleus, and tibialis anterior. We used this reduced muscle set since we only simulated sagittal-plane exoskeleton devices and since fewer muscles kept the optimizations tractable. To create the set of nine muscles, we combined muscles (from the model of [30]) that had similar sagittal-plane functions into one muscle with a combined maximum isometric force value. Joint

and muscle kinematics and net joint moments were computed through inverse kinematics and inverse dynamics tools using OpenSim 3.3 [32].

## Simulation framework

We used a simulation framework [33] based on the GPOPS-II direct collocation optimal control software (Version 2.3) [34] to solve the muscle redundancy problem for unassisted walking. In each simulation, we solved for muscle activity while enforcing muscle activation and tendon compliance dynamics. Muscle kinematics were constrained to match muscle-tendon lengths and velocities obtained from inverse kinematics, and muscle-generated moments were constrained to match net joint moments computed from inverse dynamics. Since we only included sagittal-plane muscles in our model, only sagittal-plane joint moments (hip flexion-extension, knee flexion-extension, and ankle plantarflexion-dorsiflexion) were matched in each optimization. We assumed left-right symmetry of walking and therefore only solved for muscle activity in the right leg. Each problem included reserve torque actuators in addition to muscle-generated moments to help ensure dynamic consistency; these actuators were penalized in the objective function such that the muscles were the primary actuators enforcing the joint moment constraints. Each optimal control problem was solved with the Legendre-Gauss-Radau quadrature collocation method provided by GPOPS-II using an initial mesh of 100 mesh intervals per second. The initial mesh was updated using mesh refinement with a tolerance of $10^{-3}$ to reduce muscle activation and tendon compliance dynamic errors in the solution trajectories. The resulting nonlinear programs produced from the collocation method were solved with a convergence tolerance of $10^{-3}$ using IPOPT, the non-linear optimization solver [35].

## Muscle parameter calibration

We calibrated the model's muscle parameters so that estimated muscle activations would better match EMG measurements. Our model calibration approach consisted of three main steps. In the first step, we scaled maximum isometric force values based on a previously reported relationship between muscle volume and total body mass [36]. In the second step, we optimized optimal fiber lengths, tendon slack lengths, and passive muscle strain parameters while minimizing the error between model and reported experimental passive muscle moments [37]. We used MATLAB's fmincon to minimize passive moment errors across a range of static joint positions with a rigid-tendon assumption for computing passive muscle force. In addition to the cost term penalizing deviations from experimental passive muscle moments, secondary cost terms were included to minimize total muscle passive force and prevent deviations from default parameter values which would lead to undesirable solutions with large passive forces in individual muscles.

The third step of our model calibration used EMG data to further adjust the model's muscle parameters. Passive muscle strain parameters were fixed to the values obtained from the first calibration step, and tendon slack length and optimal fiber lengths were again optimized within 25% of their original values, using the first-step calibration values as an initial guess. The error between EMG data and muscle excitations was the primary term minimized in the objective function. Passive muscle forces were also minimized to prevent undesired increases in passive forces due to the readjusted parameters. The muscle activations were also included as a lower-weighted, secondary objective term to aid convergence. The resulting muscle parameters were used in all subsequent simulations.

## Exoskeleton device simulations

After calibrating the model for a given subject, we simulated unassisted and assisted gait using the subject's remaining two gait cycles. In both unassisted and assisted gait, the primary objective was to minimize metabolic cost computed from a version of the metabolic energy model developed by Umberger et al. (2003) that was modified to have a continuous first derivative for gradient-based optimization [38, 39]. We included additional secondary objective terms to minimize muscle excitation, muscle activation, and the derivative of tendon force, all of which aided problem convergence. Since our simulation method relied on kinematics obtained from an inverse kinematics solution, the unassisted and assisted simulations used the same healthy walking kinematics (i.e., the simulation did not change the model's kinematics in response to the assistive device). In the unassisted simulations, the muscles and the heavily-penalized reserve torque actuators were the only actuators available to reproduce the net joint moments.

In the assisted simulations, exoskeleton devices were modeled as massless torque actuators and could apply torques to reduce muscle effort, while still matching the net joint moment constraints from inverse dynamics. The actuators had no power limits, but had peak torque limits for hip flexion-extension (1.0 N-m/kg), knee flexion-extension (1.0 N-m/kg), and ankle plantarflexion (2.0 N-m/kg); these peak torque limits were included to speed convergence and were chosen such that optimized device controls never exceeded the optimization bounds. Torques were applied in the following five joint directions: hip flexion, hip extension, knee flexion, knee extension, and ankle plantarflexion. Single-joint exoskeleton devices provided assistive torques in one of the five joint directions. Multi-joint exoskeleton devices provided assistance to the following combinations of joint directions: (1) hip-extension knee-extension, (2) hip-flexion knee-flexion, (3) knee-flexion ankle-plantarflexion, (4) hip-flexion ankle-plantarflexion, and (5) hip-flexion knee-flexion ankle-plantarflexion. The multi-joint exoskeleton devices were actuated by individual control signals (i.e., "independent" control) or with only one control signal applied to all joint directions (i.e., "coupled" control). When using coupled control, additional "gain" variables scaled the applied exoskeleton torques to allow different applied torque magnitudes since net joint moment magnitudes differed between the hip, knee, and ankle.

For all unassisted and assisted conditions, we computed both total and muscle-level metrics of metabolic cost to assess device performance. The *gross average total metabolic rate* was computed by integrating the sum of individual muscle metabolic rates, multiplying by two (since we only solved for the right leg and assumed medio-lateral symmetry), dividing by the motion duration and total body mass, and adding a constant basal rate of 1.2 W/kg [38]. The *average muscle metabolic rate* was computed by integrating the metabolic rate of a muscle, multiplying by two, and dividing by the duration of the motion and body mass. Changes in both gross average total metabolic rate and average muscle metabolic rate due to assistance were computed as a percent of unassisted gross average total metabolic rate.

## Validation approach

To validate our simulations, we compared musculoskeletal model outputs to experimental data. We compared the value and timing of peak joint moments and joint angles computed with OpenSim to values previously reported in the literature. Simulated muscle activations were compared to normalized EMG based on onset and offset timings as suggested by Hicks et al. (2015) [40]. For these comparisons, we defined muscles, both simulated and experimental, when their activation was above 5% of peak activation. Errors in muscle timing were defined when the simulated muscle activations were above the 5% threshold and the EMG was not above the threshold, and vice versa. We accounted for electromechanical delay in muscles

by shifting the simulated muscle activations in time by 75 ms [41]. Timing errors were computed across the gait cycle, where 0% error indicated a perfect match at all time points and 100% error indicated no match across all time points.

In addition to comparisons to experimental data, we computed a set of error metrics also based on suggestions by Hicks et al. (2015). We computed the RMS errors between experimental and model marker trajectories from inverse kinematics. To estimate the dynamic consistency of our simulations, we computed pelvis residual forces and moments from inverse dynamics across simulation gait cycles. Finally, we computed the RMS magnitude of the reserve torques to ensure that the constraints imposed to match experimental net joint moments was achieved primarily by muscle-generated torques.

## Sensitivity analysis

We performed a sensitivity analysis to ensure that the convergence tolerance used in our walking optimizations did not affect our results. We varied the convergence tolerance between 1 and $10^{-4}$ and solved the unassisted walking problem for all subjects with the same gait cycles used to generate our results. We normalized objective values using the solution generated with the $10^{-4}$ tolerance and computed the mean and standard deviation across subjects and gait cycles (S4 Fig). The objective values for the $10^{-3}$ convergence tolerance, were close to a normalized objective value of 1 in our sensitivity analysis, meaning that tightening the tolerance to $10^{-4}$ would yield no improvement in objective values. Therefore, we used a $10^{-3}$ convergence tolerance for each walking optimization in this study.

## Statistical testing

To compare the effect of devices on percent changes in metabolic cost, we employed a linear mixed model (fixed effect: device; random effect: subject) with analysis of variance (ANOVA) tests and Tukey post-hoc pairwise tests [42]. We used a significance level of $\alpha = 0.05$. The data for the statistical analyses consisted of 75 observations (5 subjects and 15 devices); we averaged over the 2 walking trials used to simulate each single and multi-joint device to remove hierarchical structure from our data [43]. The statistical tests were performed with R [44–46].

# Results

## Device performance

All 15 ideal assistance devices–single joint, multi-joint coupled, and multi-joint independent–significantly decreased average total metabolic rate compared to unassisted walking (Fig 1, S2 and S3 Tables; p < 0.05). The largest reduction in metabolic cost among multi-joint devices was produced by the hip-flexion knee-flexion ankle-plantarflexion devices (34% coupled, 39% independent). Other multi-joint devices produced large metabolic savings: hip-flexion ankle-plantarflexion (29% coupled, 34% independent), knee-flexion ankle-plantarflexion assistance (30% coupled, 32% independent), and hip-extension knee-extension assistance (12% coupled, 14% independent). While independent assistance outperformed coupled assistance, the differences between coupled and independent were small (the percent change in cost for coupled assistance was 3.5% lower on average across multi-joint devices). The single-joint hip-flexion device provided the largest savings of the single joint devices (22% reduction), closely followed by knee-flexion assistance (21%). Multi-joint devices provided greater savings compared to single joint devices for all conditions (Tukey post-hoc test, p < 0.05) except for two conditions. First, coupled and independent multi-joint hip-extension knee-extension assistance was not significantly different from single-joint hip-flexion and knee-flexion assistance. Second,

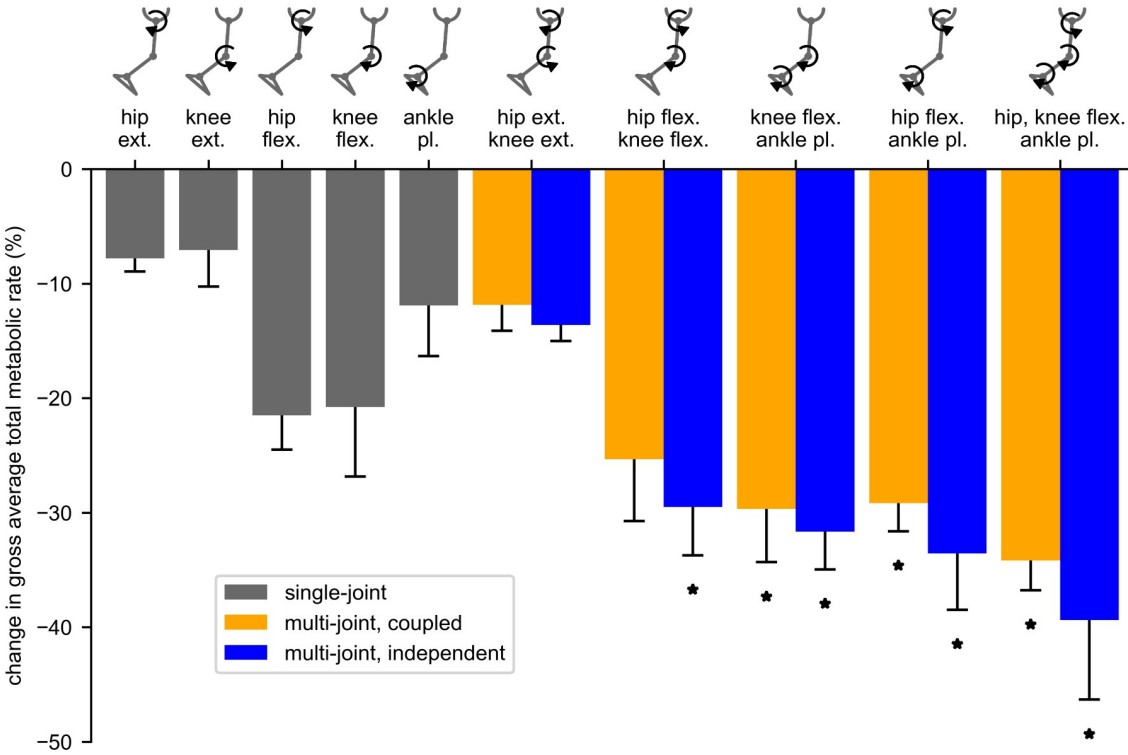

**Fig 1. Reduction in metabolic rate for single and multi-joint assistance devices.** The percent change in gross total metabolic rate, averaged over the gait cycle, for the single joint (gray), multi-joint coupled (orange), and multi-joint independent (blue) assistance devices. Negative values indicate decreases in metabolic cost. Each bar value and corresponding error bar provides the mean reduction and standard deviation across subjects. Asterisks indicate devices that produced significantly larger metabolic reductions compared to single-joint devices.

coupled hip-flexion knee-flexion assistance was not significantly different from single-joint knee-flexion assistance.

## Muscle metabolic changes

The change in average muscle metabolic rates for a given multi-joint device were similar between the coupled and independent control devices. Both coupled and independent multi-joint hip-extension knee-extension assistance produced metabolic reductions in the gluteus maximus (5% coupled, 6% independent) and vastus intermedius (6% coupled and independent) muscles (Fig 2). Multi-joint hip-flexion knee-flexion assistance primarily reduced the iliopsoas average metabolic rate (19% coupled, 20% independent), and produced smaller reductions in the gastrocnemius (4% coupled and independent) and semimembranosus (2% coupled and independent) (Fig 3). Multi-joint knee-flexion ankle-plantarflexion assistance reduced the average metabolic rates of the soleus (11% coupled, 14% independent) and gastrocnemius (4% coupled and independent), but also produced a large reduction in the iliopsoas (15% coupled and independent), which was not directly assisted (Fig 4). Iliopsoas effort was reduced since rectus femoris activity increased to counteract knee-flexion assistive torque, as seen by the small increase in rectus femoris average metabolic rate (4% coupled and independent). Multi-joint hip-flexion ankle-plantarflexion assistance produced large metabolic reductions in the iliopsoas (19% coupled and independent) and soleus (12% coupled, 14% independent) (Fig 5). Multi-joint hip-flexion knee-flexion ankle-plantarflexion assistance

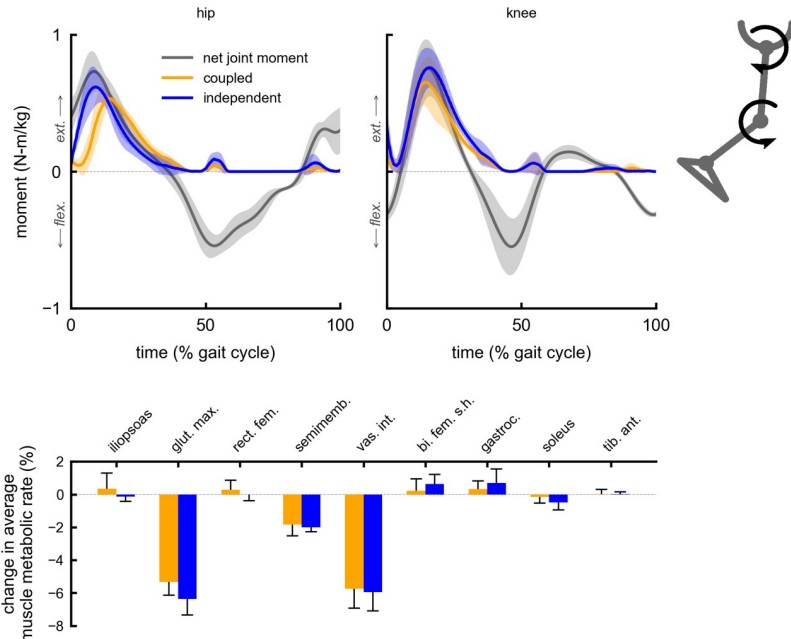

**Fig 2. Summary of multi-joint hip-extension knee-extension assistance.** Top: the device torques for multi-joint hip-extension knee-extension assistance with coupled (orange) and independent (blue) control compared to net joint moments (gray). Bottom: changes in average muscle metabolic rates as a percent of unassisted gross average total metabolic rate for the multi-joint assistive devices. Negative values indicate decreases in metabolic cost. Solid bars and error bars indicate the mean and standard deviation across subjects, respectively. Summing the individual muscle percent changes yields the total percent changes for the hip-extension knee-extension multi-joint devices reported in Fig 1.

similarly produced large iliopsoas (18% coupled, 19% independent) and soleus (11% coupled, 13% independent) metabolic reductions, and the added knee-flexion torque produced a reduction (rather than increase) in the semimembranosus (2% coupled and independent) and a larger reduction in the gastrocnemius (4% coupled and independent) (Fig 6).

## Device torques and powers

Average peak device torques and powers were similar between coupled and independent multi-joint assistance for many of the devices (S2 Table), but there were some notable differences between peak torques and powers at individual degrees-of-freedom (S4 Table). For multi-joint hip-flexion knee-flexion assistance, a lower average peak knee-flexion torque was observed with coupled control (0.4 N-m/kg) compared to independent control (0.7 N-m/kg). However, despite this peak moment decrease, coupled control provided larger knee-flexion average peak power (1.7 W/kg) compared to independent control (1.0 W/kg). The largest differences in peak device torques were seen in ankle-plantarflexion assistance for multi-joint devices, but this did not necessarily result in similarly large metabolic changes. For example, the average peak ankle plantarflexion torque for independent knee-flexion ankle-plantarflexion assistance (1.6 N-m/kg) was larger than the average peak torque for coupled assistance (1.0 N-m/kg), but these devices produced similar metabolic savings (S2 Table, Fig 4). This could be partially explained by the relatively small difference in average peak powers at the ankle for knee-flexion ankle-plantarflexion multi-joint assistance between independent (3.4 W/kg) and coupled control (3.2 W/kg) (S4 Table). These results suggest that multi-joint assistance can exploit the timing of torque assistance to provide device powers necessary for large metabolic

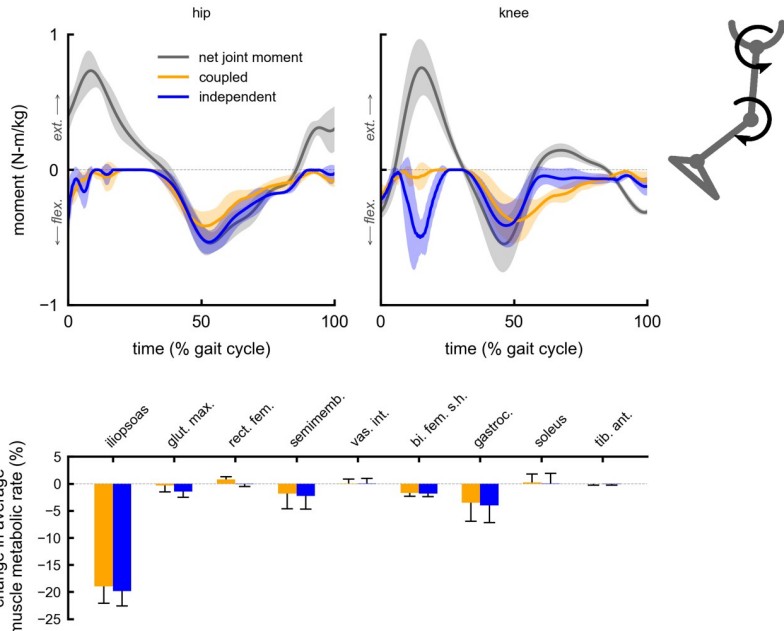

**Fig 3. Summary of multi-joint hip-flexion knee-flexion assistance.** Top: the device torques for multi-joint hip-flexion knee-flexion assistance with coupled (orange) and independent (blue) control compared to net joint moments (gray). Bottom: changes in average muscle metabolic rates as a percent of unassisted gross average total metabolic rate for the multi-joint assistive devices. Negative values indicate decreases in metabolic cost. Solid bars and error bars indicate the mean and standard deviation across subjects, respectively. Summing the individual muscle percent changes yields the total percent changes for the hip-flexion knee-flexion multi-joint devices reported in Fig 1.

savings, even when coupled torque timing limits assistance torque magnitudes at individual joints.

## Validation results

Joint moments (S1 Fig) and joint angles (S2 Fig) computed with OpenSim had similar peak values and timings compared to previously reported joint moments and joint angles for normal treadmill walking at a similar walking speed [47]. The onset-offset timing errors between simulated muscle activations and normalized EMG recordings, averaged across gait cycles and subjects, were as follows: gluteus maximus (28.4%), rectus femoris (31.4%), semimembranosus (32.1%), vastus intermedius (11.1%), gastrocnemius (17.0%), soleus (7.9%), and tibialis anterior (25.1%) (S3 Fig).

Estimates of gross total metabolic rate (3.2 ± 0.2 W/kg, S1 Table) were lower than typical experimental values for normal unassisted walking (4.0–4.3 W/kg, [48]). This metabolic underestimation is likely because we did not include frontal-plane muscles (e.g., hip adductors-abductors) or upper-extremity muscles in our musculoskeletal model. However, since we were evaluating trends in percent metabolic changes between sagittal-plane single-joint and multi-joint devices and between coupled and independent multi-joint devices rather than absolute values of metabolic cost, we deemed this underestimation acceptable for the purposes of this study.

The RMS errors between experimental and model marker trajectories from inverse kinematics had a mean value of 2.2 cm across lower-limb markers and simulation gait cycles. The mean RMS error in the pelvis residual forces, expressed as a percent of the peak ground reaction force (GRF) magnitude, was 4.8%, which is within the 5% guideline suggested by Hicks

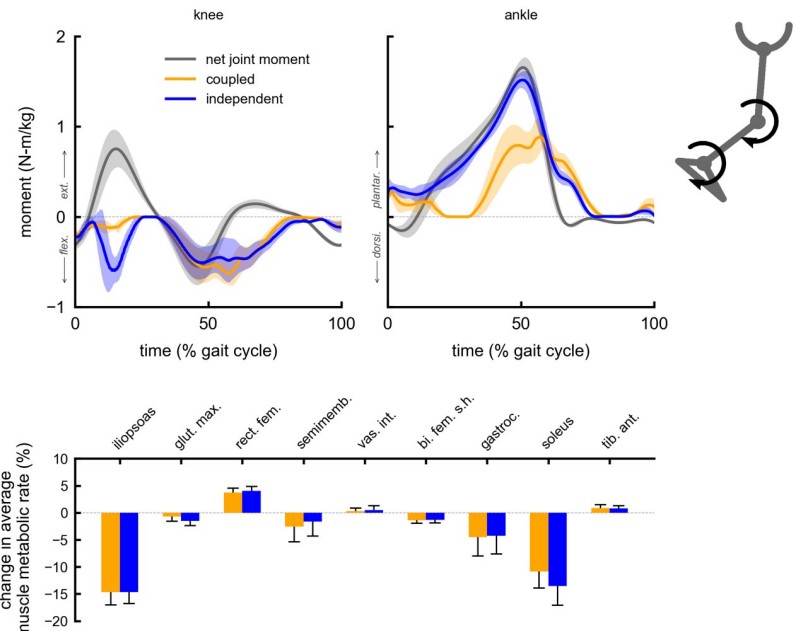

**Fig 4. Summary of multi-joint knee-flexion ankle-plantarflexion assistance.** Top: the device torques for multi-joint knee-flexion ankle-plantarflexion assistance with coupled (orange) and independent (blue) control compared to net joint moments (gray). Bottom: changes in average muscle metabolic rates as a percent of unassisted gross average total metabolic rate for the multi-joint assistive devices. Negative values indicate decreases in metabolic cost. Solid bars and error bars indicate the mean and standard deviation across subjects, respectively. Summing the individual muscle percent changes yields the total percent changes for the knee-flexion ankle-plantarflexion multi-joint devices reported in Fig 1.

et al. (2015). The mean RMS error in the pelvis residual moments, expressed as a percentage of the product of average mass center height and peak GRF magnitude, was 2.8%, which exceeds the 1% guideline suggested by Hicks et al. (2015). However, given the good agreement between net joint moments and previously reported walking data and between muscle activity and EMG measurements, we deemed this error to be acceptable. Finally, the RMS magnitude of the reserve torques had a mean value of 0.06 N-m across degrees of freedom and simulations; the maximum error across time, degrees of freedom, and simulations was 2.01 N-m. The ratio of the RMS reserve magnitude to the maximum absolute net joint moment had mean and peak values of 0.1% and 3.7%, respectively, which meet the guideline of 5% provided by Hicks et al. (2015).

## Discussion

We found that most multi-joint devices, both coupled and independent, could provide significantly larger metabolic savings compared to single-joint torque assistance in simulated lower-limb exoskeleton devices for walking. This is noteworthy considering that most current exoskeleton devices only assist one degree-of-freedom [3, 5, 8, 11, 49–51]. This is also promising for the recent development of multi-joint exoskeletons [12, 14, 15]. Our results suggests that designers should consider coupled multi-joint assistance when building multi-joint exoskeletons since coupled devices could reduce device weight and simplify device architecture by requiring fewer actuators. In addition, the largest metabolic reduction with coupled assistance occurred with hip-flexion knee-flexion ankle-plantarflexion assistance, suggesting that assisting more than two joints with one actuator can be beneficial. In many cases, peak assistive

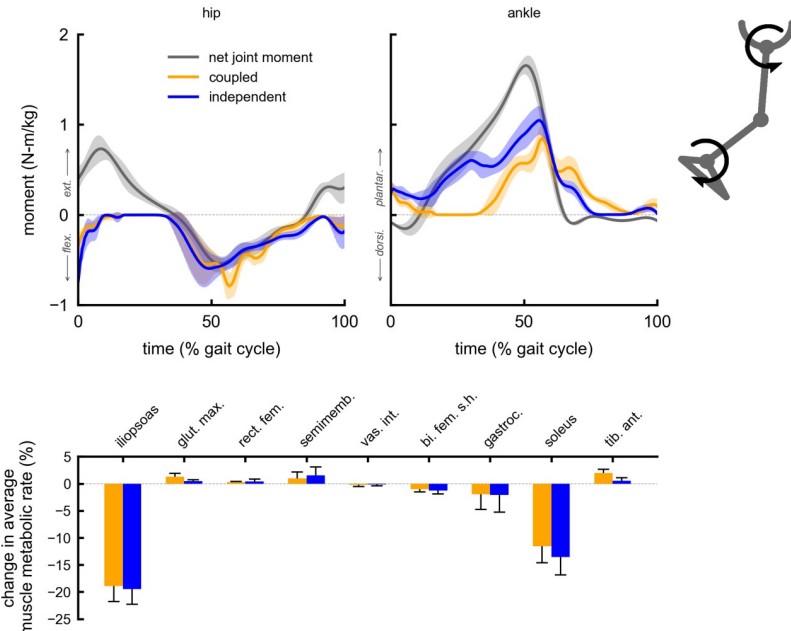

**Fig 5. Summary of multi-joint hip-flexion ankle-plantarflexion assistance.** Top: the device torques for multi-joint hip-flexion ankle-plantarflexion assistance with coupled (orange) and independent (blue) control compared to net joint moments (gray). Bottom: changes in average muscle metabolic rates as a percent of unassisted gross average total metabolic rate for the multi-joint assistive devices. Negative values indicate decreases in metabolic cost. Solid bars and error bars indicate the mean and standard deviation across subjects, respectively. Summing the individual muscle percent changes yields the total percent changes for the hip-flexion ankle-plantarflexion multi-joint devices reported in Fig 1.

moments and powers were similar between coupled and independent multi-joint assistive devices, in spite of the reduced control complexity when coupling assistance between joints.

We did not model device masses in our simulations, which would increase metabolic cost estimates, especially when adding mass to distal body segments [52]. We chose to assess the benefit from torque assistance separately from the exoskeleton designs, since devices that apply the same assistance can have varying metabolic penalties depending on actuator torque and power densities. This approach is similar to that of exoskeleton emulator systems, which use off-board motors to deliver torque assistance to the user and eliminate the cost of worn masses from actuators. In addition, when implementing our simulated assistance strategies in experiments, designers can account for the metabolic cost for wearing a particular exoskeleton design using the mass distribution of the device (e.g., by using the relationships in Browning et al. (2007) [52]).

Other limitations of our simulation approach should be considered when interpreting our results. As previously mentioned, we excluded frontal plane muscles (e.g., hip adductors-abductors) from our simulations, but these muscles have important functions in walking, and this could partially explain why our simulation underestimates whole-body metabolic cost relative to ranges reported in the literature (S1 Table). Since muscles that act in the sagittal plane often also have moment arms in the frontal plane (e.g., adductor longus), our simulations may exclude muscle interactions between sagittal and frontal plane degrees of freedom [26]. We also did not include upper-extremity muscles in our simulations, which would have contributed to our total metabolic cost estimates. Since we used a minimal muscle set in our musculoskeletal model, absolute predictions of metabolic cost would be less reliable than comparisons between simulated assistance conditions. Therefore, for this study, we focused on the

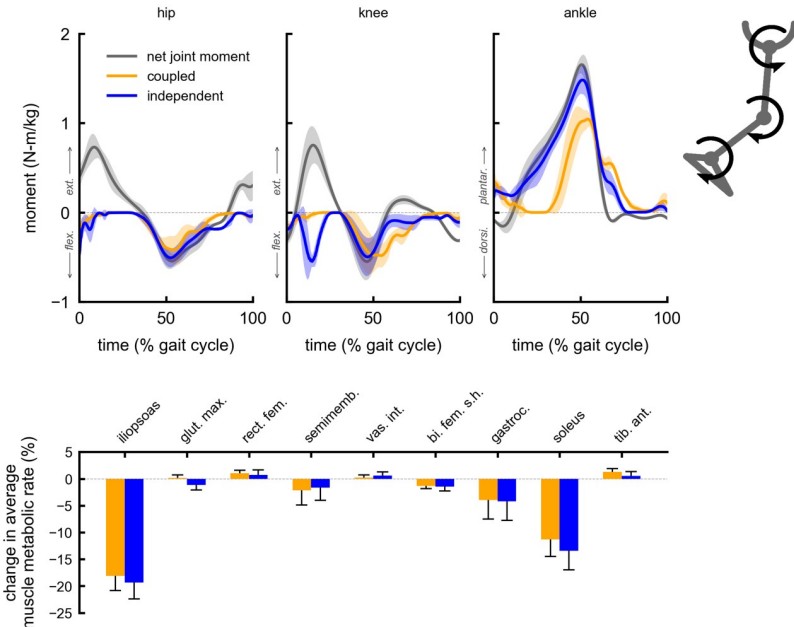

**Fig 6. Summary of multi-joint hip-flexion knee-flexion ankle-plantarflexion assistance.** Top: the device torques for multi-joint hip-flexion knee-flexion ankle-plantarflexion assistance with coupled (orange) and independent (blue) control compared to net joint moments (gray). Bottom: changes in average muscle metabolic rates as a percent of unassisted gross average total metabolic rate for the multi-joint assistive devices. Negative values indicate decreases in metabolic cost. Solid bars and error bars indicate the mean and standard deviation across subjects, respectively. Summing the individual muscle percent changes yields the total percent changes for the hip-flexion knee-flexion ankle-plantarflexion multi-joint devices reported in Fig 1.

metabolic trends between single-joint and multi-joint devices and between coupled and independent multi-joint devices. As previously mentioned, there were quantitative differences between simulated and measured muscle activity (S3 Fig), so uncertainty in muscle model model parameters obtained from our calibration step may have also contributed to metabolic cost underestimates. In addition, while we optimized for metabolic cost in our simulations, users in exoskeleton experiments would likely also consider comfort, balance, or joint injury risk in response to assistance, and these factors may affect metabolic cost measurements. Absolute metabolic cost predictions from simulation could be made more accurate by including a full whole-body muscle set, including upper-extremity muscles, and optimizing for user comfort and safety (e.g., minimizing joint contact forces and ligament strains). Finally, we created simulations using experimental gait data from only five subjects, which may partially explain why some of the multi-joint devices we tested did not produce significantly different metabolic cost changes compared to single-joint devices.

It is important to address how the noted limitations of the study are relevant to designers aiming to test coupled control strategies in experiments. First, we excluded frontal plane muscles in our simulations since we only tested exoskeleton devices acting in the sagittal plane. Some hip abductor-adductor muscles produce moments in the sagittal plane; thus, we would likely see differences in some of the metabolic changes we observed had we included these muscles. However, these frontal plane muscles would likely also benefit from device assistance, potentially leading to overall greater reductions. We also did not model upper extremity muscles, but including these muscles in our simulation approach would shift metabolic reductions for all devices by the same amount and not change the trends we observed. Lastly, we did not model device masses, which could influence percent metabolic changes as device masses

increase. However, as noted previously, we chose to separate the effect of assistance from device mechanical design. Therefore, we would expect the trends we observed to hold in experiments where worn masses are consistent across tested assistance strategies. In summary, we believe that the metabolic trends observed between single-joint and multi-joint devices and between coupled and independent assistance will be replicable in experiments, even if differences in absolute metabolic cost measurements are observed.

Future studies should build upon the simulation methods used in this study to further improve metabolic predictions. Users in experiments often adapt their kinematics in response to assistance (e.g., [5, 11, 12, 20, 27, 53–55]), but our simulations utilized an approach where kinematics were prescribed exactly based on normal walking data. Devices may cause different changes in walking kinematics depending on which joints were assisted and the torque or power applied to the user. Therefore, the metabolic cost trends we observed in our simulations could differ depending on the magnitude of kinematic adaptations between single and multi-joint devices. Predictive simulation methods that can optimize kinematic changes in addition to muscle adaptations could provide a better understanding of why exoskeleton users often change their gait with assistance. The inclusion of muscle synergies to constrain muscle activation predictions has been shown to improve predictions of subject-specific walking motions [56] and could potentially improve predictions of user adaptations to exoskeleton assistance. In addition, it has been shown that personalizing joint axes, electromechanical delays, activation dynamics time constants, and other musculoskeletal parameters can affect metabolic cost estimates and should be considered for future calibration methods [57]. Finally, muscle kinematic states estimated from ultrasound measurements for both assisted and unassisted walking could be used to calibrate metabolic models and improve predictions [58].

Future work should include experimental testing of assistance strategies designed through simulation to help reveal where simulation methods fall short. For example, our group recently successfully reduced the metabolic cost of walking for a hip-knee-ankle exoskeleton using simulation-designed assistance [28], but percent changes in metabolic cost and estimated muscle activity changes from the simulation did not match well with experimental measurements. Therefore, combining simulations and experiments in an iterative loop could be particularly effective for designing assistive devices to reduce metabolic cost. Experiments should test the multi-joint strategies we simulated in this study to verify the metabolic relationships between coupled and independent control strategies and should especially consider coupled hip-flexion knee-flexion ankle-plantarflexion assistance, since this device outperformed all the two-joint devices in our simulations. Simulations could pair with experiments in other novel ways aside from this "predict-test-validate" framework. With the advent of human-in-the-loop optimization methods, simulation may not need to predict metabolic cost changes with high accuracy to have utility, but only to generate good initial guesses or help optimizers prioritize promising assistance control strategies.

## Conclusion

We used musculoskeletal modeling and optimal control methods to simulate 15 single-joint and multi-joint ideal assistance devices. This work helps provide an understanding of the musculoskeletal factors driving the metabolic benefits of multi-joint assistance. Our results, showing that the greatest reduction in metabolic cost using a single actuator to assist multiple joints (39% reduction) was significantly larger than the reduction produced by the best single-joint device (22% reduction), suggest that exoskeleton designers should consider coupled assistance when designing multi-joint devices. Coupled assistance approaches could simplify wearable devices, increase metabolic reductions when actuation is limited, and help keep experiment

times tractable. Designers can use these results as a guide for generating new hypotheses to test in exoskeleton experiments or when prototyping new exoskeleton designs. We invite researchers to use our freely available data (https://simtk.org/projects/coupled-exo-sim) and code (https://github.com/stanfordnmbl/coupled-exo-sim) to build upon our work.

## Supporting information

**S1 Fig. Net joint moments.** Left: net joint moments from inverse dynamics for individual subjects. Right: joint moment means (black) and standard deviations (gray bands) across subjects.
(TIF)

**S2 Fig. Joint angles.** Left: joint angles from inverse kinematics for individual subjects. Right: joint angle means (black) and standard deviations (gray bands) across subjects.
(TIF)

**S3 Fig. Experimental electromyography data compared to unassisted simulation activations.** This figure shows electromyography data (gray bands) recorded from walking experiments compared to optimized activations generated from unassisted simulations (black). Both electromyography data and simulated activations are averaged across gait cycles not included in the muscle parameter calibration procedure.
(TIF)

**S4 Fig. Sensitivity of objective value to convergence tolerance.** The mean (bars) and standard deviation (whiskers) of normalized objective values for unassisted walking solutions across subjects and gait cycles. Objective values at each convergence tolerance are normalized by objective values using a convergence tolerance of $10^{-4}$. We used a convergence tolerance of $10^{-3}$ to generate our results, since tightening the tolerance to $10^{-4}$ had little effect on the objective (i.e., the normalized objective values were close to one for the $10^{-3}$ tolerance).
(TIF)

**S5 Fig. Muscle activations for unassisted and assisted simulations.** This figure shows muscle activations averaged across subjects for unassisted walking (black), single-joint assisted walking (gray), and multi-joint coupled (orange) and independent (blue) assisted walking.
(TIF)

**S1 Table. Simulation-predicted unassisted metabolic rates.** This table shows the predicted gross average total metabolic rates for each subject. The columns represent the gait cycles used when testing single and multi-joint devices. These values underestimate experimental values typical of normal unassisted walking (4.0–4.3 W/kg, [48]).
(TIF)

**S2 Table. Total metabolic reductions and device powers.** This table shows (a) relative and (b) absolute reductions in gross average total metabolic rate and the (c) peak positive, (d) average positive, and (e) average negative power for each single and multi-joint device. Quantities in columns (b)-(e) are normalized by subject mass. All columns are reported as mean ± standard deviation across 5 subjects.
(TIF)

**S3 Table. Subject-specific relative metabolic reductions.** This table shows subject-specific relative reductions in gross average total metabolic rate for each single and multi-joint device. All quantities are percent reductions in metabolic cost relative to unassisted walking.
(TIF)

**S4 Table. Peak device moments and powers at each degree-of-freedom.** This table shows the peak device moments and powers for individual degrees-of-freedom for each single and multi-joint device. All quantities are normalized by subject mass and are reported as mean ± standard deviation across 5 subjects. Peak moment values are peak magnitudes of device moments applied at each degree-of-freedom.
(TIF)

## Acknowledgments

We thank Brendan Quinlivan, Ye Ding, and Conor Walsh of the Harvard Biodesign Lab and Gwen Bryan and Steve Collins of the Stanford Biomechatronics Lab for discussing experimental multi-joint exoskeleton designs which helped inspire this study.

## Author Contributions

**Conceptualization:** Nicholas A. Bianco, Patrick W. Franks, Jennifer L. Hicks, Scott L. Delp.

**Data curation:** Nicholas A. Bianco.

**Formal analysis:** Nicholas A. Bianco.

**Funding acquisition:** Jennifer L. Hicks, Scott L. Delp.

**Investigation:** Nicholas A. Bianco.

**Methodology:** Nicholas A. Bianco.

**Project administration:** Jennifer L. Hicks, Scott L. Delp.

**Software:** Nicholas A. Bianco.

**Supervision:** Jennifer L. Hicks, Scott L. Delp.

**Validation:** Nicholas A. Bianco, Patrick W. Franks, Jennifer L. Hicks.

**Visualization:** Nicholas A. Bianco.

**Writing – original draft:** Nicholas A. Bianco, Patrick W. Franks, Jennifer L. Hicks.

**Writing – review & editing:** Nicholas A. Bianco, Patrick W. Franks, Jennifer L. Hicks, Scott L. Delp.

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
