## [Decision Letter · Decision Letter 0]

5 Aug 2021

PONE-D-21-10817

Coupled exoskeleton assistance simplifies control and maintains metabolic benefits: a simulation study

PLOS ONE

Dear Dr. Bianco,

Thank you for submitting your manuscript to PLOS ONE. After careful consideration, we feel that it has merit but does not fully meet PLOS ONE’s publication criteria as it currently stands. Therefore, we invite you to submit a revised version of the manuscript that addresses the points raised during the review process.

Thank you for this submission. The reviewers acknowledged the importance of the proposed research and pointed to the clarity of this manuscript. They brought up a couple of major questions. The questions related to the assumptions and limitations in this study (reviewer 1) and the methodological questions related to the statistical support of claims (reviewer 2) should be adequately addressed.

We look forward to receiving your revised manuscript.

Kind regards,

Sergiy Yakovenko

Academic Editor

PLOS ONE

Journal Requirements:

Additional Editor Comments (if provided):

Thank you for this submission. The reviewers acknowledged the importance of the proposed research and pointed to the clarity of this manuscript. They brought up a couple of major questions. The questions related to the assumptions and limitations in this study (reviewer 1) and the methodological questions related to the statistical support of claims (reviewer 2) should be adequately addressed.

Reviewers' comments:

Reviewer's Responses to Questions

**Comments to the Author**

1. Is the manuscript technically sound, and do the data support the conclusions?

Reviewer #1: Partly

Reviewer #2: Partly

2. Has the statistical analysis been performed appropriately and rigorously? 

Reviewer #1: N/A

Reviewer #2: No

3. Have the authors made all data underlying the findings in their manuscript fully available?

Reviewer #1: Yes

Reviewer #2: Yes

4. Is the manuscript presented in an intelligible fashion and written in standard English?

Reviewer #1: Yes

Reviewer #2: Yes

5. Review Comments to the Author

Reviewer #1: Summary:

This paper aims at investigating the relative benefits of exoskeleton assistance applied at lower limb joint(s). Musculoskeletal simulations are well employed to explore the design space possibilities of a coupled control algorithm. The results demonstrate the potential of coupled control of multi-joint exoskeletons to yield a metabolic benefit. The manuscript is well-written and the results of the manuscript may guide future development of assistive devices in this field. There are several issues to address.

Major Comments

1. While I understand the rationale behind using ideal massless actuators for simulation in this research, it is possible this assumption could affect the generalization of these results toward real world implementation of exoskeletons. Specifically, adding the mass of an assistive device to more distal joints would result in larger increases in lower limb moment of inertia and may have a larger metabolic penalty than adding the same mass to proximal joints. In this way, it’s possible that the added mass at a joint or multiple joints could affect the relative metabolic benefit of applied assistance as simulated here.

2. The term ' whole-body metabolic rate' is misleading in reference to simulated metabolic rate because the authors are using a metabolic probe that incorporates muscle activity on a model with limited muscles in the lower limb, and no upper limb simulated muscle activity. The lack of the upper limb activity is not mentioned as a limitation, nor its potential effect on relative metabolic performance across simulated conditions. On Line 184 a reference should also be provided for the 1.2 W/kg basal rate.

3. The author includes language in the methods/results section that would more appropriately be in the discussion. This is especially true in the "Comparison of simulations with experimental results" section, which may be more appropriate as a subsection of results rather than methods. Specifically, any comparison of the presented results to existing literature (lines 209-210), or interpretation of results (e.g. lines 211-212, 296-299) should be relocated to the discussion.

4. The authors compare the metabolic savings of exoskeletons in literature to the results of simulations presented in the manuscript (lines 315-336) and offer many reasons why the simulated metabolic benefits are larger than measured metabolic rate. We agree with the authors’ assertion in lines 212-216 that the metabolic quantity calculated for this work is sufficient for comparing percent metabolic changes between assisted/unassisted simulations; however, there are several limitations to comparing the simulated metabolic rate to metabolic rates reported in literature which should be addressed: (1) The authors did not record any experimental metabolic measurements, and are using the minimization of simulated metabolic rate in the optimization, so there is no verification of the accuracy of simulated metabolic rate with experimental data (2) the calculation of simulated metabolic cost here excludes upper limb muscles and several lower limb muscles (3) the referenced previously collected data was limited to lower limb kinematics, and therefore the metabolic impacts of upper limb kinematics including trunk swing and arm motion were excluded.

5. The authors are correct that the use of massless idealized actuators may impact the comparison of metabolic rates with experimental studies compared to the study by Quinlivan et al. (2017) (line 322). However, rather than only acknowledging the impacts of added mass on an individual comparison of simulated vs experimental metabolic outcomes, a statement at the beginning or end of this paragraph that references the metabolic impact of added mass effects on the simulations themselves and their relative performance should be added.

6. The authors acknowledge that no kinematic changes were permitted between simulated conditions. However, additional discussion of whether different combinations of assistance are more of less likely to elicit altered kinematics, and how that may impact results.

Minor

- Lines 22-23 remove the word "from"

-Lines 34-38 this statement is a bit difficult/unclear to read, especially with the use of "either" twice

- Line 39 define the metric of 'success' referenced

- Line 43-45 the sentence is unclear and contractions should be expanded

- Line 46 remove the word still

- Line 74-75 missing the word "compared" before "to"

- Line 263 Muscle metabolic changes section could use quantitative values in the text to contextualize the stated reductions.

Reviewer #2: The proposed manuscript is a computational study of the potential benefits of multi- and coupled-joint actuated exoskeletons. The study design is well conceived and straight-forward with reasonable modeling assumptions and could provide useful insight into the design of exoskeletons. However, there are several significant issues that must be addressed. Specifically, the manuscript lacks appropriate statistical analyses and does not provide sufficient subject-specific data. These limitations, combined with a relatively small sample size (5 participants, 3 gait cycles per participant), make it difficult to evaluate the study’s conclusions and could undermine the findings. These issues are described in more depth below.

MAJOR REVISIONS

METHODS

Currently, the study lacks any inferential statistics or hypothesis testing. Although the paper makes two specific claims, 1) that multi-joint assistance increases metabolic savings compared to single-joint assistance and 2) that coupled multi-joint assistance achieves similar metabolic savings to single-joint assistance, neither of these hypotheses are specifically tested. This is particularly worrisome with the modest sample size used. For example, Figure 1 shows changes in gross average whole-body metabolic rate. The manuscript claims:

Lines 259-261: “Multi-joint devices provided greater savings compared to single joint devices for all conditions except for multi-joint hip-extension knee-extension assistance, which was outperformed by single-joint hip-flexion and knee-flexion assistance.”

While it is true that the average savings were greater for multi-joint devices, the error bars in Figure 1 are nontrivial. Appropriate hypothesis tests should be performed, especially with such a limited sampling size. Furthermore, the data would be more transparent for the reader if individual subject values and/or variances were provided in the main text and figures. While many of these raw data values are provided in the supplementary data, their omission from the primary manuscript could facilitate misinterpretation. The combination of 1) small sample size, 2) insufficient statistical methods, 3) frequent reliance on averaged values, and 4) unforthcoming individual values make the conclusions difficult to evaluate and could undermine readers’ confidence in the study findings. Therefore, it is critical that these issues be addressed across all the results and figures.

Another specific example can be found in the section titled ‘Comparison of simulations with experimental results’:

Lines 194-195: “The simulated muscle activations were similar to normalized EMG with a few exceptions (S3 Fig).”

This language is very obtuse and subjective. Supplementary Figure 3 shows average recorded and simulated EMG profiles, but no quantification of their similarity. Some examples of error are sparsely listed:

Lines 202-207: “The average peak values of simulated soleus and gastrocnemius activity were within 7% and 5%, respectively of the EMG measurements, but peaks occurred 13% and 9% later in the gait cycle, respectively, compared to the EMG measurements. Average peak simulated tibialis anterior activity was similar to the peak timing of EMG measurements (within 6% of the gait cycle), but had differences in activity magnitudes for some subjects”

However, it is not clear how these errors are calculated, e.g. RMSE. Nor does it provide an indication of the variability of these errors across muscles or participants. Cross correlation, regression, or normalized RMSE would all provide better clarity and transparency of the model accuracy and one of these metrics, or an appropriate alternative, should be performed for each muscle.

DISCUSSION

Overall, the discussion is well written and clear. The authors give reasonable speculation about why their simulations may have overestimated metabolic changes and, importantly, acknowledge several limitations of their work. They also appropriately relate their findings to other studies in the field of exoskeletons.There are, however, several claims which can not yet be made until the aforementioned issues are addressed and appropriate hypothesis tests are performed. They include:

Lines 301-303: “We found that multi-joint torque assistance could provide larger metabolic savings compared to single-joint torque assistance in simulated lower-limb exoskeleton devices for walking.”

Lines 306-309: “We found that the simulated multi-joint exoskeletons using coupled torque assistance could provide similar metabolic savings to those using independently-controlled torque assistance. This result suggests that exoskeleton designers should consider coupling torque actuators when building multi-joint exoskeletons.”

MINOR REVISIONS

INTRODUCTION

Line 1: “Wearable robotic exoskeletons that reduce the metabolic cost of walking could improve mobility for individuals with musculoskeletal or neurological impairments and assist soldiers and firefighters carrying heavy loads.”

The current phrasing of this sentence insinuates that exoskeletons ONLY help soldiers and firefighters but their applications in the general population are much broader.

Line 34: “Coupled assistance could greatly simplify the mechanical and control design of exoskeleton devices either by reducing either the number of actuators needed for a device or by simplifying control complexity (i.e., the number of parameters personalized to a subject) and thus reducing the time needed to perform human-in-the-loop optimizations to achieve good reductions in metabolic cost.”

I believe there is a typo here: “…either by reducing either…”. There should be only one ‘either’.

6. PLOS authors have the option to publish the peer review history of their article (what does this mean?). If published, this will include your full peer review and any attached files.

Reviewer #1: No

Reviewer #2: No

---

## [Author Response · Author response to Decision Letter 0]

21 Sep 2021

Reviewer #1 

Summary

This paper aims at investigating the relative benefits of exoskeleton assistance applied at lower limb joint(s). Musculoskeletal simulations are well employed to explore the design space possibilities of a coupled control algorithm. The results demonstrate the potential of coupled control of multi-joint exoskeletons to yield a metabolic benefit. The manuscript is well-written and the results of the manuscript may guide future development of assistive devices in this field. There are several issues to address.

We are glad the reviewer finds that our manuscript has the potential to positively impact future exoskeleton development via our coupled control approach. We are grateful for the reviewer’s thoughtful comments, which have improved the manuscript. We have revised the manuscript to address the reviewers comments, as described below.

Major Comments

1. While I understand the rationale behind using ideal massless actuators for simulation in this research, it is possible this assumption could affect the generalization of these results toward real world implementation of exoskeletons. Specifically, adding the mass of an assistive device to more distal joints would result in larger increases in lower limb moment of inertia and may have a larger metabolic penalty than adding the same mass to proximal joints. In this way, it’s possible that the added mass at a joint or multiple joints could affect the relative metabolic benefit of applied assistance as simulated here.

We agree that the mass added to a body segment can influence the net metabolic change produced by an assistive device, especially when mass is added to distal body segments (Browning et al. 2007). We chose to exclude the masses of actuators from our devices to directly evaluate how the choice of control strategy affects changes in metabolic cost. This is because there could be multiple exoskeleton devices capable of applying a particular type of assistance, but the metabolic penalty of the device would depend on the device's design, mass-efficiency, and actuator torque and power densities. These designers could then estimate the expected mass penalty for their particular design using the mass distribution of the device. This approach is also often used experimentally by exoskeleton researchers. Exoskeleton experiments sometimes use emulator systems to reduce the mass added to the user by using off-board motors when designing device control strategies (e.g., Zhang et al. 2017, Quinlivan et al. 2017). With emulator systems, the expected benefit of assistance can be assessed independent of device architecture so that exoskeleton designers could know what benefit to expect. In this way, our approach mimics the approach of emulator experiments. We also excluded the constant mass of the emulator, since the metabolic cost of wearing this mass is also constant across control conditions. Adding this constant cost to our simulations would change percent differences in metabolic cost, but would not change the trends in metabolic cost changes we observed in our simulations.

We have added the following paragraph to the Discussion section to clarify how excluding device masses may affect the performance of each simulated device and how this may impact our results:

We did not model device masses in our simulations, which would increase metabolic cost estimates, especially when adding mass to distal body segments (Browning et al. 2007). We chose to assess the benefit from torque assistance separately from the exoskeleton designs, since devices that apply the same assistance can have varying metabolic penalties depending on mass-efficiency and actuator torque and power densities. This approach is similar to that of exoskeleton emulator systems, which use off-board motors to deliver torque assistance to the user and eliminate the cost of worn masses from actuators. While emulator systems add mass to the user, this mass, and the resulting metabolic cost, is constant across conditions. Therefore, while including the mass from an emulator system in our simulations would increase absolute metabolic cost predictions, it would likely not affect the trends in metabolic cost changes we observed in our experiments. In addition, when implementing our simulated assistance strategies in experiments, designers can account for the metabolic cost for wearing a particular exoskeleton design using the mass distribution of the device (e.g., by using the relationships in Browning et al. (2007)). 

2. The term ' whole-body metabolic rate' is misleading in reference to simulated metabolic rate because the authors are using a metabolic probe that incorporates muscle activity on a model with limited muscles in the lower limb, and no upper limb simulated muscle activity. The lack of the upper limb activity is not mentioned as a limitation, nor its potential effect on relative metabolic performance across simulated conditions. On Line 184 a reference should also be provided for the 1.2 W/kg basal rate.

We agree that the term “whole-body metabolic rate” is misleading due to the lack of upper extremity muscles. We’ve renamed this term to “lower-limb metabolic rate” in the text and in the figures to better reflect the metabolic quantity we computed. We have added the following text to the Discussion section to explain this limitation (line 370):

We also did not include upper extremity muscles in our simulations, which would have contributed to our total metabolic cost estimates.

We have also revised the sentence starting on line 382:

...could be made more accurate by including a whole-body muscle set, including upper-extremity muscles, and optimizing for user comfort...

We have also added the reference for the 1.2 W/kg basal metabolic rate (Umberger et al. 2003) on line 190.

3. The author includes language in the methods/results section that would more appropriately be in the discussion. This is especially true in the "Comparison of simulations with experimental results" section, which may be more appropriate as a subsection of results rather than methods. Specifically, any comparison of the presented results to existing literature (lines 209-210), or interpretation of results (e.g. lines 211-212, 296-299) should be relocated to the discussion.

We agree that much of the content in the subsection “Comparison of simulations with experimental results” is appropriate within the Results. We have moved the appropriate text from this subsection to the Results and added a paragraph to the Methods summarizing our validation approach. While the lines comparing the validation results to existing literature (original manuscript lines 209-210, 211-212, and 296-299) could potentially be moved to the Discussion, we have left them in the Results section for clarity.

4. The authors compare the metabolic savings of exoskeletons in literature to the results of simulations presented in the manuscript (lines 315-336) and offer many reasons why the simulated metabolic benefits are larger than measured metabolic rate. We agree with the authors’ assertion in lines 212-216 that the metabolic quantity calculated for this work is sufficient for comparing percent metabolic changes between assisted/unassisted simulations; however, there are several limitations to comparing the simulated metabolic rate to metabolic rates reported in literature which should be addressed: (1) The authors did not record any experimental metabolic measurements, and are using the minimization of simulated metabolic rate in the optimization, so there is no verification of the accuracy of simulated metabolic rate with experimental data (2) the calculation of simulated metabolic cost here excludes upper limb muscles and several lower limb muscles (3) the referenced previously collected data was limited to lower limb kinematics, and therefore the metabolic impacts of upper limb kinematics including trunk swing and arm motion were excluded.

The reviewer raises many good questions about our comparisons between experimental metabolic cost reductions and the metabolic changes we observed from our simulation study. After considering these points, we agree that these comparisons are not as useful as the comparisons between simulation conditions due to the assumptions made for our study. We have decided to remove this paragraph and instead expand the Discussion to address the other comments made by the reviewer.

5. The authors are correct that the use of massless idealized actuators may impact the comparison of metabolic rates with experimental studies compared to the study by Quinlivan et al. (2017) (line 322). However, rather than only acknowledging the impacts of added mass on an individual comparison of simulated vs experimental metabolic outcomes, a statement at the beginning or end of this paragraph that references the metabolic impact of added mass effects on the simulations themselves and their relative performance should be added.

We agree that we should acknowledge if excluding device masses on our comparisons between simulated devices would affect predicted metabolic savings. We excluded device masses to isolate the effect of each assistance strategy independently from the variable device architectures that could deliver this torque assistance. If we were to include device masses in our simulations, we would assume a constant device architecture, similar to exoskeleton emulator experiments. A constant device architecture would add a constant mass to the exoskeleton user, which would incur a constant metabolic cost across simulation conditions. Adding this constant cost to our simulations would change percent differences in metabolic cost, but would not change the trends in metabolic cost changes we observed in our simulations. We have added the following paragraph to the discussion to summarize our modeling choices regarding device masses:

We did not model device masses in our simulations, which would increase metabolic cost estimates, especially when adding mass to distal body segments (Browning et al. 2007). We chose to assess the benefit from torque assistance separately from the exoskeleton designs, since devices that apply the same assistance can have varying metabolic penalties depending on mass-efficiency and actuator torque and power densities. This approach is similar to that of exoskeleton emulator systems, which use off-board motors to deliver torque assistance to the user and eliminate the cost of worn masses from actuators. While emulator systems add mass to the user, this mass, and the resulting metabolic cost, is constant across conditions. Therefore, while including the mass from an emulator system in our simulations would increase absolute metabolic cost predictions, it would likely not affect the trends in metabolic cost changes we observed in our experiments. In addition, when implementing our simulated assistance strategies in experiments, designers can account for the metabolic cost for wearing a particular exoskeleton design using the mass distribution of the device (e.g., by using the relationships in Browning et al. (2007)). 

6. The authors acknowledge that no kinematic changes were permitted between simulated conditions. However, additional discussion of whether different combinations of assistance are more of less likely to elicit altered kinematics, and how that may impact results.

The reviewer raises an important point that the fixed kinematics assumption may affect simulated devices differently depending which joints are assisted and the number of joints assisted. We have added the following to the Discussion paragraph starting on line 390 to provide more details about this modeling assumption and how it may have impacted our results:

Devices may cause different changes in walking kinematics depending on which joints were assisted and the torque or power applied to the user. Therefore, the metabolic cost trends we observed in our simulations could differ depending on the magnitude of kinematic adaptations between single and multi-joint devices.

Minor Comments

- Lines 22-23 remove the word "from"

We have made this change.

- Lines 34-38 this statement is a bit difficult/unclear to read, especially with the use of "either" twice

We have improved the clarity of the sentence on these lines by rephrasing to the following:

Coupled assistance could simplify the control design of exoskeleton devices by reducing control complexity (i.e., the number of parameters personalized to a subject) and thus reducing the time needed to perform human-in-the-loop optimizations to achieve desired reductions in metabolic cost. Coupled assistance could also simplify the mechanical design of exoskeletons by reducing the number of actuators needed for a device which could be lighter and impose less restriction on the user.

- Line 39 define the metric of 'success' referenced

We have clarified that “success” in this sentence refers to metabolic cost reductions:

Assisting two joints at once using one actuator, or “coupling" assistance, produced significant reductions in metabolic cost in recent exoskeleton studies with an ankle-hip soft exosuit [12, 19-21] and a knee-ankle device [14].

- Line 43-45 the sentence is unclear and contractions should be expanded

We have rephrased the sentence on these lines to be clearer:

Other exoskeletons that assist multiple joints may be effective, but they have not yet been tested in experiments, since optimizing controls for multiple joints is often resource-intensive.

- Line 46 remove the word still

We have made this change.

- Line 74-75 missing the word "compared" before "to"

We have made this change.

- Line 263 Muscle metabolic changes section could use quantitative values in the text to contextualize the stated reductions.

We have added quantitative values for the muscle metabolic changes in the results subsection. We have also added error bars in the bar charts for Figures 2 through 6 representing standard deviations in muscle metabolic reductions across subjects.

Reviewer #2 

The proposed manuscript is a computational study of the potential benefits of multi- and coupled-joint actuated exoskeletons. The study design is well conceived and straight-forward with reasonable modeling assumptions and could provide useful insight into the design of exoskeletons. However, there are several significant issues that must be addressed. Specifically, the manuscript lacks appropriate statistical analyses and does not provide sufficient subject-specific data. These limitations, combined with a relatively small sample size (5 participants, 3 gait cycles per participant), make it difficult to evaluate the study’s conclusions and could undermine the findings. These issues are described in more depth below.

We are thankful for the reviewer’s thoughtful comments and are glad that our study shows promise for providing insight into device design. We have added statistical tests, which have strengthened our results and study conclusions, and provided better quantitative comparisons between simulated and experimental data. The revised manuscript addresses the reviewer’s comments as described below.

Major Comments

METHODS

Currently, the study lacks any inferential statistics or hypothesis testing. Although the paper makes two specific claims, 1) that multi-joint assistance increases metabolic savings compared to single-joint assistance and 2) that coupled multi-joint assistance achieves similar metabolic savings to single-joint assistance, neither of these hypotheses are specifically tested. This is particularly worrisome with the modest sample size used. For example, Figure 1 shows changes in gross average whole-body metabolic rate. The manuscript claims:

Lines 259-261: “Multi-joint devices provided greater savings compared to single joint devices for all conditions except for multi-joint hip-extension knee-extension assistance, which was outperformed by single-joint hip-flexion and knee-flexion assistance.”

While it is true that the average savings were greater for multi-joint devices, the error bars in Figure 1 are nontrivial. Appropriate hypothesis tests should be performed, especially with such a limited sampling size. Furthermore, the data would be more transparent for the reader if individual subject values and/or variances were provided in the main text and figures. While many of these raw data values are provided in the supplementary data, their omission from the primary manuscript could facilitate misinterpretation. The combination of 1) small sample size, 2) insufficient statistical methods, 3) frequent reliance on averaged values, and 4) unforthcoming individual values make the conclusions difficult to evaluate and could undermine readers’ confidence in the study findings. Therefore, it is critical that these issues be addressed across all the results and figures.

We agree with the reviewer that statistical analyses would provide more confidence in our results. We performed statistical tests and found that the metabolic changes from multi-joint devices were significantly different from those from single joint devices (Tukey post-hoc test, p < 0.05), with the following exceptions:

Coupled hip-flexion, knee-flexion assistance was not significantly different from knee-flexion only assistance.

Coupled hip-extension, knee-extension assistance was not significantly different from hip-extension only and knee-extension only assistance.

Independent hip-extension, knee-extension assistance was not significantly different from hip-extension only assistance

We have added the following paragraph to the Methods section to describe our statistical testing:

To compare the effect of devices on percent changes in metabolic cost, we employed a linear mixed model (fixed effect: device; random effect: subject) with analysis of variance (ANOVA) tests and Tukey post-hoc pairwise tests (Bretz et al., 2011). We used a significance level of α = 0.05. The data for the statistical analyses consisted of 75 observations (5 subjects and 15 devices); we averaged over the 2 walking trials used to simulate each single and multi-joint device to remove hierarchical structure from our data (Samuels et al., 1999). The statistical tests were performed with R (Core Team R, 2021; Bates et al., 2015; Hothorn et al., 2008).

We have revised the Results section to include our findings from our statistical testing. Starting on line 237:

All 15 ideal assistance devices–single joint, multi-joint coupled, and multi-joint independent–significantly decreased average whole-body metabolic rate compared to unassisted walking (Fig 1, S6 Table, S7 Table; p < 0.05).

Starting on line 249:

Multi-joint devices provided greater savings compared to single joint devices for all conditions (Tukey post-hoc test, p < 0.05) except for two conditions. First, coupled and independent multi-joint hip-extension knee-extension assistance was not significantly different from single-joint hip-flexion and knee-flexion assistance. Second, coupled hip-flexion knee-flexion assistance was not significantly different from single-joint knee-flexion assistance. 

Finally, we have added a new table in the supplementary material (S7 Table) that includes subject-specific metabolic reductions across all devices, and we have cited this table in the main text.

Another specific example can be found in the section titled ‘Comparison of simulations with experimental results’:

Lines 194-195: “The simulated muscle activations were similar to normalized EMG with a few exceptions (S3 Fig).”

This language is very obtuse and subjective. Supplementary Figure 3 shows average recorded and simulated EMG profiles, but no quantification of their similarity. Some examples of error are sparsely listed:

Lines 202-207: “The average peak values of simulated soleus and gastrocnemius activity were within 7% and 5%, respectively of the EMG measurements, but peaks occurred 13% and 9% later in the gait cycle, respectively, compared to the EMG measurements. Average peak simulated tibialis anterior activity was similar to the peak timing of EMG measurements (within 6% of the gait cycle), but had differences in activity magnitudes for some subjects”

However, it is not clear how these errors are calculated, e.g. RMSE. Nor does it provide an indication of the variability of these errors across muscles or participants. Cross correlation, regression, or normalized RMSE would all provide better clarity and transparency of the model accuracy and one of these metrics, or an appropriate alternative, should be performed for each muscle.

We agree that the comparisons between predicted muscle activations and experimental EMG signals could be improved. We computed a new metric to quantify the error in the onset and offset timings between simulated muscles and the EMG signals based on the suggestion provided by Hicks et al. (2015). We defined muscles, both simulated and experimental, as activated when above 5% of peak activation; this activation threshold was chosen to only compare regions of significant muscle activity. Errors in muscle timing were defined when the simulated muscle activations were above the 5% threshold and the EMG was not above the threshold, and vice versa. We accounted for electromechanical delay in muscles by shifting the simulated muscle activations in time by 75 ms (Seth and Pandy, 2007). Timing errors were computed across the gait cycle, where 0% error indicated a perfect match at all time points and 100% error indicated no match across all time points. The timing errors, averaged across gait cycles and subjects, were as follows: gluteus maximus (28.4%), rectus femoris (31.4%), semimembranosus (32.1%), vastus intermedius (11.1%), gastrocnemius (17.0%), soleus (7.9%), and tibialis anterior (25.1%).

We’ve updated the sections in the Methods and Results to describe these new quantitative comparisons between muscle activations and EMG signals.

DISCUSSION

Overall, the discussion is well written and clear. The authors give reasonable speculation about why their simulations may have overestimated metabolic changes and, importantly, acknowledge several limitations of their work. They also appropriately relate their findings to other studies in the field of exoskeletons.There are, however, several claims which can not yet be made until the aforementioned issues are addressed and appropriate hypothesis tests are performed. They include:

Lines 301-303: “We found that multi-joint torque assistance could provide larger metabolic savings compared to single-joint torque assistance in simulated lower-limb exoskeleton devices for walking.”

Lines 306-309: “We found that the simulated multi-joint exoskeletons using coupled torque assistance could provide similar metabolic savings to those using independently-controlled torque assistance. This result suggests that exoskeleton designers should consider coupling torque actuators when building multi-joint exoskeletons.”

We are glad that the Discussion section is clear, and we agree that the conclusions could be strengthened by the suggested hypothesis testing. By addressing the comments related to statistical testing above, we believe we have sufficiently supported these conclusions by showing that most multi-joint devices (both using independent and coupled control) produced significantly greater metabolic cost savings compared to single-joint devices. We have rephrased the conclusion on lines 336-340 to reflect the result from our statistical testing:

We found that for most multi-joint devices, the metabolic savings achieved with both coupled and independent torque control were significantly greater compared to single-joint devices. This result suggests that designers should consider coupled multi-joint assistance when building multi-joint exoskeletons, especially when reducing the number of actuators in an exoskeleton can optimize device weight and architecture.

Minor Comments

INTRODUCTION

Line 1: “Wearable robotic exoskeletons that reduce the metabolic cost of walking could improve mobility for individuals with musculoskeletal or neurological impairments and assist soldiers and firefighters carrying heavy loads.”

The current phrasing of this sentence insinuates that exoskeletons ONLY help soldiers and firefighters but their applications in the general population are much broader.

We have rephrased this line to imply that exoskeletons could have a positive impact on populations outside of the examples we provide:

Wearable robotic exoskeletons that reduce the metabolic cost of walking could improve mobility for many individuals including those with musculoskeletal or neurological impairments and soldiers and firefighters who frequently carry heavy loads.

Line 34: “Coupled assistance could greatly simplify the mechanical and control design of exoskeleton devices either by reducing either the number of actuators needed for a device or by simplifying control complexity (i.e., the number of parameters personalized to a subject) and thus reducing the time needed to perform human-in-the-loop optimizations to achieve good reductions in metabolic cost.”

I believe there is a typo here: “…either by reducing either…”. There should be only one ‘either’.

We have fixed this typo and improved the clarity of this sentence on by rephrasing it to the following:

Coupled assistance could greatly simplify the control design of exoskeleton devices by reducing control complexity (i.e., the number of parameters personalized to a subject) and thus reducing the time needed to perform human-in-the-loop optimizations to achieve reductions in metabolic cost. Coupled assistance could also simplify the mechanical design of exoskeletons by reducing the number of actuators needed for a device.

---

## [Decision Letter · Decision Letter 1]

20 Oct 2021

PONE-D-21-10817R1Coupled exoskeleton assistance simplifies control and maintains metabolic benefits: a simulation studyPLOS ONE

Dear Dr. Bianco,

Thank you for submitting your manuscript to PLOS ONE. After careful consideration, we feel that it has merit but does not fully meet PLOS ONE’s publication criteria as it currently stands. Therefore, we invite you to submit a revised version of the manuscript that addresses the points raised during the review process.

Thank you for the revision that addressed all the major questions, as indicated by the enthusiastic review by both reviewers. There are a couple of minor points leftover that would improve the quality of this publication. The discussion could potentially have a brief interpretation of the null results and the requested discussion expansion of limitations. It would also be useful to identify the significant differences in the plotted comparisons. Otherwise, this submission is ready for publication.

We look forward to receiving your revised manuscript.

Kind regards,

Sergiy Yakovenko

Academic Editor

PLOS ONE

Journal Requirements:

Additional Editor Comments (if provided):

Thank you for the revision that addressed all the major questions. There are a couple of minor points leftover that would improve the quality of this publication. The discussion could potentially have a brief interpretation of the null results and the requested discussion expansion of limitations. It would also be useful to identify the significant differences in the plotted comparisons. Otherwise, this submission is ready for publication.

Reviewers' comments:

Reviewer's Responses to Questions

**Comments to the Author**

1. If the authors have adequately addressed your comments raised in a previous round of review and you feel that this manuscript is now acceptable for publication, you may indicate that here to bypass the “Comments to the Author” section, enter your conflict of interest statement in the “Confidential to Editor” section, and submit your "Accept" recommendation.

Reviewer #1: All comments have been addressed

Reviewer #2: All comments have been addressed

2. Is the manuscript technically sound, and do the data support the conclusions?

Reviewer #1: Yes

Reviewer #2: Yes

3. Has the statistical analysis been performed appropriately and rigorously? 

Reviewer #1: Yes

Reviewer #2: Yes

4. Have the authors made all data underlying the findings in their manuscript fully available?

Reviewer #1: Yes

Reviewer #2: Yes

5. Is the manuscript presented in an intelligible fashion and written in standard English?

Reviewer #1: Yes

Reviewer #2: Yes

6. Review Comments to the Author

Reviewer #1: (No Response)

Reviewer #2: The authors have appropriately addressed my previous comments regarding the manuscript titled ”Coupled exoskeleton assistance simplifies control and maintains metabolic benefits: a simulation study”. While I am satisfied with the edits, there are some minor points that I think could be expanded upon which would increase the overall quality of the publication. These points are described below.

Methods

After performing the requested statistical testing, there are several comparisons in which the null hypothesis could not be rejected. It would be worthwhile for the authors to speculate why these specific instances did not provide metabolic savings compared to the other coupled assistance systems. Could insufficient sampling be excluded as a possibility? Or is 5 participants and only 3 steps of locomotion imply insufficient to detect the metabolic savings? If the sampling is appropriate, why do only some of the coupled systems show metabolic savings?

Discussion

The authors have, very appropriately, acknowledged several limitations, e.g. massless actuators, of their simulation study and have emphasized the necessity of experimental data to validate their findings. It may be worthwhile for the authors to expand on why they think these findings will be validated or why the assumptions they made are reasonable.

Minor Issues

It would be very helpful to the readers to indicate on the figures which comparisons were statistically significant (with * or some other visual).

7. PLOS authors have the option to publish the peer review history of their article (what does this mean?). If published, this will include your full peer review and any attached files.

Reviewer #1: No

Reviewer #2: No

---

## [Author Response · Author response to Decision Letter 1]

15 Nov 2021

Reviewer #2 

The authors have appropriately addressed my previous comments regarding the manuscript titled ”Coupled exoskeleton assistance simplifies control and maintains metabolic benefits: a simulation study”. While I am satisfied with the edits, there are some minor points that I think could be expanded upon which would increase the overall quality of the publication. These points are described below.

We are glad that our edits have addressed the reviewer’s major concerns. We have further revised the manuscript to address the reviewer’s remaining comments related to statistical testing and the value of this study in light of the limitations, as described in detail below. 

Methods

After performing the requested statistical testing, there are several comparisons in which the null hypothesis could not be rejected. It would be worthwhile for the authors to speculate why these specific instances did not provide metabolic savings compared to the other coupled assistance systems. Could insufficient sampling be excluded as a possibility? Or is 5 participants and only 3 steps of locomotion imply insufficient to detect the metabolic savings? If the sampling is appropriate, why do only some of the coupled systems show metabolic savings?

We agree that insufficient sampling could explain why the null hypothesis could not be rejected for these devices. Accordingly, we have added a sentence on lines 368-371:

Finally, we created simulations using experimental gait data from only five subjects, which may partially explain why some of the multi-joint devices we tested did not produce significantly different metabolic cost changes compared to single-joint devices.

We chose not to speculate broadly on why certain multi-joint devices did not produce significantly different metabolic cost changes compared to single-joint devices.

Discussion

The authors have, very appropriately, acknowledged several limitations, e.g. massless actuators, of their simulation study and have emphasized the necessity of experimental data to validate their findings. It may be worthwhile for the authors to expand on why they think these findings will be validated or why the assumptions they made are reasonable.

We are glad that the previous revisions adequately address the reviewer’s comments about the limitations of our study. As suggested, we updated the discussion to note why our findings are valid and assumptions are reasonable. These updates are included in the revised manuscript (line 372-388).

Minor Issues

It would be very helpful to the readers to indicate on the figures which comparisons were statistically significant (with * or some other visual).

As suggested, we have updated Figure 1 to include asterisks to visualize which multi-joint devices produced significantly different metabolic cost changes compared to their respective single-joint devices. We have also updated the Figure 1 caption to reflect the change.

---

## [Decision Letter · Decision Letter 2]

1 Dec 2021

Coupled exoskeleton assistance simplifies control and maintains metabolic benefits: a simulation study

PONE-D-21-10817R2

Dear Dr. Bianco,

We’re pleased to inform you that your manuscript has been judged scientifically suitable for publication and will be formally accepted for publication once it meets all outstanding technical requirements.

Kind regards,

Sergiy Yakovenko

Academic Editor

PLOS ONE

Additional Editor Comments (optional):

Thank you for the thorough revision. All reviewers are in agreement that this work is ready for publication.

Reviewers' comments:

Reviewer's Responses to Questions

**Comments to the Author**

1. If the authors have adequately addressed your comments raised in a previous round of review and you feel that this manuscript is now acceptable for publication, you may indicate that here to bypass the “Comments to the Author” section, enter your conflict of interest statement in the “Confidential to Editor” section, and submit your "Accept" recommendation.

Reviewer #2: All comments have been addressed

2. Is the manuscript technically sound, and do the data support the conclusions?

Reviewer #2: Yes

3. Has the statistical analysis been performed appropriately and rigorously? 

Reviewer #2: Yes

4. Have the authors made all data underlying the findings in their manuscript fully available?

Reviewer #2: Yes

5. Is the manuscript presented in an intelligible fashion and written in standard English?

Reviewer #2: Yes

6. Review Comments to the Author

Reviewer #2: The authors have addressed all of my previous concerns. They have added an appropriate statistical analysis, acknowledged the study's limitations, and provided reasonable rationale for their modeling assumptions. Although the sample size could still be a reason for pause, this limitation has been acknowledge more explicitly.

7. PLOS authors have the option to publish the peer review history of their article (what does this mean?). If published, this will include your full peer review and any attached files.

Reviewer #2: No

---

## [Editor Report · Acceptance letter]

14 Dec 2021

PONE-D-21-10817R2 

Coupled exoskeleton assistance simplifies control and maintains metabolic benefits: a simulation study 

Dear Dr. Bianco:

I'm pleased to inform you that your manuscript has been deemed suitable for publication in PLOS ONE. Congratulations! Your manuscript is now with our production department. 

Kind regards, 

on behalf of

Dr. Sergiy Yakovenko 

Academic Editor

PLOS ONE